# LHDNN: Maintaining High Precision and Low Latency Inference of Deep Neural Networks on Encrypted Data

**Jiaming Qian [1], Ping Zhang [1,2,*], Haoyong Zhu [1], Muhua Liu [1], Jiechang Wang [3] and Xuerui Ma [1]**

1  School of Mathematics and Statistics, Henan University of Science and Technology, Luoyang 471023, China; jmqian@stu.haust.edu.cn (J.Q.); 201410040135@stu.haust.edu.cn (H.Z.); lxk0379@126.com (M.L.); 201410050112@stu.haust.edu.cn (X.M.)
2  Intelligent System Science and Technology Innovation Center, Longmen Laboratory, Luoyang 471023, China
3  Sports Big Data Center, Department of Physical Education, Zhengzhou University, Zhengzhou 450001, China; wangjiechang@126.com
*  Correspondence: zping@haust.edu.cn

**Abstract:** The advancement of deep neural networks (DNNs) has prompted many cloud service providers to offer deep learning as a service (DLaaS) to users across various application domains. However, in current DLaaS prediction systems, users' data are at risk of leakage. Homomorphic encryption allows operations to be performed on ciphertext without decryption, which can be applied to DLaaS to ensure users' data privacy. However, mainstream homomorphic encryption schemes only support homomorphic addition and multiplication, and do not support the ReLU activation function commonly used in the activation layers of DNNs. Previous work used approximate polynomials to replace the ReLU activation function, but the DNNs they implemented either had low inference accuracy or high inference latency. In order to achieve low inference latency of DNNs on encrypted data while ensuring inference accuracy, we propose a low-degree Hermite deep neural network framework (called LHDNN), which uses a set of low-degree trainable Hermite polynomials (called LotHps) as activation layers of DNNs. Additionally, LHDNN integrates a novel weight initialization and regularization module into the LotHps activation layer, which makes the training process of DNNs more stable and gives a stronger generalization ability. Additionally, to further improve the model accuracy, we propose a variable-weighted difference training (VDT) strategy that uses ReLU-based models to guide the training of LotHps-based models. Extensive experiments on multiple benchmark datasets validate the superiority of LHDNN in terms of inference speed and accuracy on encrypted data.

**Keywords:** DLaaS; homomorphic encryption; privacy protection; CKKS FHE scheme; deep neural networks; hermite polynomials

## 1. Introduction

Deep neural networks (DNNs) have been widely used in various areas, such as image classification, target detection, and natural language processing, due to their strong predictive abilities [1]. However, DNNs require a large amount of complex computation, and users may encounter limited computing power or lack the necessary expertise to deploy these models. As a result, many users turn to cloud services to deploy their models, giving rise to deep learning as a service (DLaaS), which allows users to access predictive services on a pay-as-you-go basis. While DLaaS offers great convenience, it also poses a potential security risk for sensitive data (e.g., medical and financial information) if cloud servers are not trusted [2,3].

We focus on using homomorphic encryption [4] to address the problem of data privacy protection in cloud services. Homomorphic encryption allows for operations to be performed directly on encrypted data, resulting in the same outcome as if the operations were performed on the plaintext. In our solution (shown in Figure 1), users encrypt their

data and upload it to the cloud server. The server then performs inference on the encrypted data without knowing the original information, protecting the user's data privacy [5]. Our solution does not reveal any data information to other users in the cloud and can be considered to have zero communication consumption. The user simply uploads data to the cloud and receives the results. Other secure multi-party computation (MPC)-based protocols require two servers to be online at the same time, which means more communication consumption [6–8].

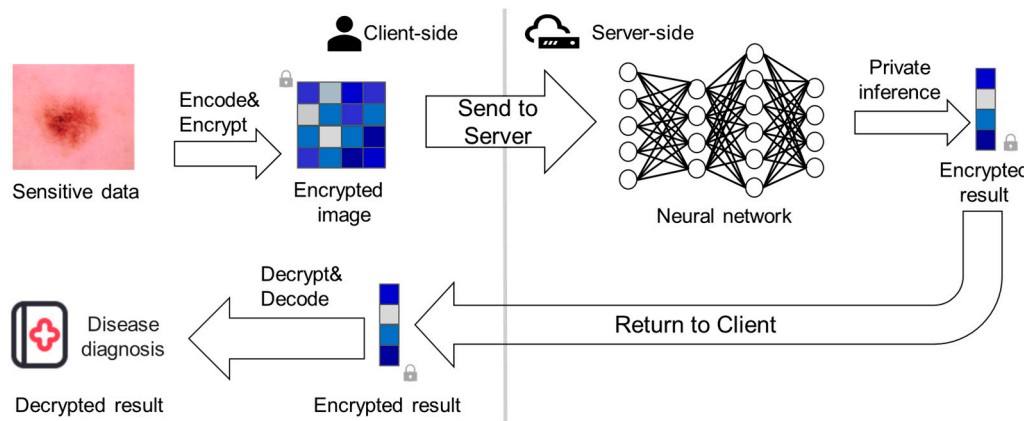

**Figure 1.** DLaaS based on homomorphic Encryption (Taking the classification of Skin-cancer Images as an example).

However, current mainstream homomorphic encryption schemes only support additive and multiplicative operations, and cannot support nonlinear activation layers in neural networks. Dowlin et al. [9] proposed CryptoNets, the first convolutional neural network (CNN) to perform privacy inference on ciphertexts of homomorphic encryption. CryptoNets use squared functions instead of standard activation functions, but this results in some accuracy degradation. Some subsequent work used polynomials obtained by approximating the standard activation function to achieve higher model accuracy, but this approach is only applicable to shallow models [10–13]. Recently, Lee et al. [14] used minimax approximate polynomials [13] to achieve an accurate approximation of the ReLU activation function, and implemented the first inference of the deep residual network ResNet-20 on encrypted data. However, its approximate polynomial degree is as high as 27, which led to extremely high inference latency. Therefore, applying homomorphic encryption to deep neural networks while maintaining both high accuracy and low inference latency has become an urgent problem.

In our paper, our goal is to reduce the inference latency of DNNs on homomorphic encrypted data while maintaining their inference accuracy. To achieve this goal, we propose a low-degree Hermite deep neural network framework (called LHDNN). Unlike existing works that use fixed-coefficient polynomials to replace the ReLU function, LHDNN uses a set of low-degree trainable Hermite polynomials (called LotHps) as the activation layers in DNNs. The degree of LotHps is two, and its low degree property ensures low inference latency for encrypted data. Specifically, LotHps are linear combinations of the first three terms of Hermite polynomials, with three trainable weight parameters that can be learned by the backpropagation algorithm during model training. Compared with the fixed coefficient polynomial, the parameterized activation function LotHps has stronger expressibility, which makes the upper limit of model accuracy higher. In addition, LHDNN combines a new weight initialization and regularization module with LotHps to ensure more stable model training and a stronger generalization ability. To further improve the accuracy of LotHps-based models, we propose a variable-weighted difference training (VDT) strategy, which uses an existing ReLU-based model to guide the training of the LotHps-based model. Specifically, the difference between the activation layer outputs of the LotHps-based model and the ReLU-based model, as well as the difference between their final layer outputs, are

added to the loss of the LotHps-based model, with a weight function p(x) to smoothly transition between the two different terms. This strategy enables the LotHps-based model to achieve higher accuracy in the early stages of training and prevents overfitting to the ReLU-based activation layer outputs, leading to lower final accuracy.

In summary, our contributions are as follows:

- We propose a low-degree Hermite deep neural network framework (called LHDNN), which employs a set of low-degree trainable Hermite polynomials (referred to as LotHps) as activation layers in the DNNs. In addition, LHDNN integrates a novel weight initialization and regularization module with LotHps, ensuring a more stable training process and a stronger model generalization ability.
- We propose a variable-weighted difference training (VDT) strategy that uses the original ReLU-based model to guide the training of the LotHps-based model, thereby improving the accuracy of the LotHps-based model.
- Our extensive experiments on benchmark datasets MNIST, Skin-Cancer, and CIFAR-10 validated the superiority of LHDNN in inference speed and accuracy on encrypted data.

In the rest of this paper, we discuss related work in Section 2. The knowledge related to homomorphic encryption is introduced in Section 3. In Section 4, we present the proposed LHDNN and the variable-weighted difference training (VDT) strategy. Relevant experiments are conducted in Sections 5 and 6. Finally, we provide a summary of the entire paper in Section 7.

## 2. Related Work

The solutions for applying homomorphic encryption to deep neural networks can be divided into two categories depending on the homomorphic encryption scheme used, i.e., homomorphic encryption schemes based on the learning with errors (LWE) puzzle and homomorphic encryption schemes based on the ring learning with errors (RLWE) puzzle.

Using the first class of homomorphic encryption solutions, nonlinear operations in the activation function can be implemented using a lookup table. While this approach can accurately evaluate the activation function in a short time, it does not support batch (SIMD) operations, leading to inefficient operations in other steps (e.g., matrix multiplication in the convolutional layer). FHE-DiNN [15] and TAPAS [16] use binarized weights and sparsification techniques to achieve faster computation on complex models, but they have a reduced inference accuracy of about 3–6.2%, even on small MNIST datasets. Lou and Jiang [17] implemented privacy inference for the ResNet-18 model on the CIFAR-10 dataset using a leveled version of the Torus homomorphic encryption (TFHE) scheme. Folkerts et al. [18] used a ternary neural network to optimize privacy-preserving inference based on TFHE. Compared to plaintext inference, it is slower by 1.7 to 2.7 orders of magnitude, but its accuracy on the MNIST dataset is only 93.1%. DOREN [19] proposed a low-depth batched neuron that can simultaneously evaluate multiple ReLU functions without approximation. The amortized runtime is about 20 times faster than Lou and Jiang's approach. Meftah et al. [20] reexamined and improved the framework proposed by DOREN, achieving a 6–34 times speedup on some CNN architectures on the CIFAR10 dataset.

The second type of homomorphic encryption scheme supports SIMD operation, i.e., packing multiple plaintexts into one ciphertext, which can significantly improve the efficiency of ciphertext operations, but their inability to support nonlinear activation functions in neural networks becomes the biggest limitation of such solutions. Dowlin et al. [9] used squared activation functions instead of standard ones to achieve inference on ciphertexts for a model with only two activation layers, achieving 98.95% accuracy on the MNIST dataset. Chabanne et al. [10] implemented a neural network with six nonlinear layers using the Taylor expansion to approximate the Softplus activation function combined with a batch normalization (BN) layer, achieving 99.30% accuracy on the MNIST dataset, slightly lower than the 99.59% of the original ReLU-based model. Hesamifard et al. [12] approximated the derivative of the ReLU activation function using a 2-degree polynomial and then replaced

the ReLU activation function with a 3-degree polynomial obtained through integration, further improving the accuracy on the MNIST dataset, but reducing the absolute accuracy by about 2.7% when used for a deeper model on the CIFAR-10 dataset. Alsaedi et al. [21] approximated the ReLU function using the Legendre polynomials and achieved a plaintext accuracy of 99.80% on the MNIST dataset, but did not evaluate their model on encrypted data. Yagyu et al. [22] improved model accuracy by pretraining the polynomial approximation coefficients of the MISH activation function. Their ciphertext accuracy on the MNIST dataset was 0.01% higher than plaintext accuracy, but their encrypted accuracy on CIFAR-10 was only 67.20%. Lee et al. [14] utilized advanced min-max approximate polynomials to achieve the best activation function approximation and successfully implemented ResNet-20 on the RNS-CKKS homomorphic encryption scheme for the first time. Although their method achieved about $92.43\% \pm 2.65\%$ inference accuracy on the CIFAR-10 dataset, the degree of their polynomial is very high, which results in a higher inference delay because ciphertext multiplication is very expensive. In addition, a large number of bootstrapping operations are needed to refresh ciphertext noise, which may cause decryption errors.

Although solutions based on the first type of homomorphic encryption have an advantage in ciphertext inference speed in the activation layers of DNN, their lack of support for batch processing results in slower inference speeds in non-activation layers. The second type of solution can achieve fast inference in non-activation layers, but currently has limited methods for handling activation layers. Using low-degree polynomials can only achieve privacy-preserving inference of encrypted data in shallow networks, but applying this method to deeper networks results in a significant decrease in model accuracy. On the other hand, using high-degree polynomials can achieve high model accuracy, but the ciphertext inference latency is very high. Therefore, efficient privacy-preserving inference of deeper DNN using FHE solutions is an important research topic that needs to be addressed. To address the limitations of current research, we propose a low-degree Hermite deep neural network framework (called LHDNN). LHDNN uses a set of low-degree trainable Hermite polynomials (referred to as LotHps) as activation layers in the DNN. LotHps contains three weight parameters that can be learned during the model training process through backpropagation algorithm. By combining a novel weight initialization and regularization module with LotHps, we can ensure a more stable training process and stronger model generalization ability. Furthermore, we propose a variable-weighted difference training (VDT) strategy that uses the original ReLU-based model to guide the training of the LotHps-based model, thereby improving the accuracy of the LotHps-based model.

## 3. Preliminaries

### 3.1. Fully Homomorphic Encryption

Homomorphic encryption has been an active area of research for over 30 years, with the first reasonably implementable fully homomorphic encryption scheme being proposed by Gentry in 2009 [4]. In this paper, we adopt the Cheon–Kim–Kim–Song (CKKS) encryption scheme proposed by Cheon et al. [23], which is considered the most suitable for machine learning applications due to its support for floating-point operations. The CKKS scheme consists of the following seven components.

- $KeyGen(\lambda)$: Given the security parameter $\lambda$, choose $M = M(\lambda, Q)$, choose an integer $P = P(\lambda, Q)$
- Sample $s \leftarrow \chi_s$, output $sk = (1, s)$.
- Sample $a \leftarrow \mathcal{R}_Q$ and $e \leftarrow \chi_e$, output $pk = \left( b = [-a \cdot s + e]_q, a \right)$.
- Sample $a' \leftarrow \mathcal{R}_{PQ}$ and $e' \leftarrow \chi_e$, output $evk = \left( b' = [-a's + e' + Ps^2]_{PQ}, a' \right)$
- $Enc(pk, m)$: For $m \in \mathcal{R}$, sample $r \leftarrow \chi_r$ and $e_0, e_1 \leftarrow \chi_e$, output $c = [r \cdot pk + (m + e_0, e_1)]_q$.
- $Dec(sk, c)$: For $c = (b, a) \in \mathcal{R}_q^2$, output $m = [b + as]_q$.
- $Add(c, c')$: Given two ciphertexts $c, c' \in \mathcal{R}_q^2$, output $c_{\text{add}} = c + c' \bmod q, c_{\text{add}} \in \mathcal{R}_q^2$.

- *Mult(c, c′)*: Given two ciphertexts $c = (b_1, a_1), c′ = (b_2, a_2) \in \mathcal{R}_q^2$, let $(d_0, d_1, d_2) = [(b_1 b_2, a_1 b_2 + a_2 b_1, a_1 a_2)]_q$, output $c_{mult} = \left[ (d_0, d_1) + \left\lfloor P^{-1} d_2 \cdot evk \right\rceil \right]_q$.
- *ReScale$_{l \to l-1}$(c)*: Given $R$ a ciphertext $c \in \mathcal{R}_{q_l}^2$ at level $l$, output $c′ = \left[ \left\lfloor \frac{q_{l-1}}{q_l} c \right\rceil \right]_{q_{l-1}}$.

The CKKS scheme uses large integers which require a high computational complexity. To reduce this complexity, Cheon et al. [24] proposed a variant of the CKKS scheme called RNS-CKKS. In this variant, large integers are split into several small integers, and the addition and multiplication operations on the original large integer are equivalent to the corresponding operations on the small integers in the residue number system.

## 3.2. Bootstrapping of CKKS

The *ReScale* operation is crucial for the homomorphic multiplication of ciphertexts. However, each *ReScale* operation reduces the modulus of the ciphertext, which means there is a limit on the number of homomorphic multiplications that can be performed. Nevertheless, bootstrapping technology can solve this problem. The bootstrapping of CKKS mainly consists of four parts: *ModRaise*, *CoeffToSlot*, *EvalMod*, and *SlotToCoeff*.

- *ModRaise*: If a ciphertext $ct$ contains the plaintext $m(X) = [\langle ct, sk \rangle]_q$, then $t(X) = \langle ct, sk \rangle = qI(X) + m(X) \equiv m(X) (\bmod q)$, where $|I(X)|_\infty < K = \mathcal{O}\left(\sqrt{h}\right)$, and h is the number of 1 in sk. The purpose of *ModRaise* is to increase the ciphertext modulus q to a large modulus Q, such that $[t(X)]_Q = t(X)$.
- *CoeffToSlot*: There is a modular reduction to be performed on the coefficients of the polynomial, but we need to approximate the modular reduction function using homomorphic addition and multiplication. Homomorphic addition and multiplication are done for the numbers in the slots, so we put the coefficients in the slots. This process is equivalent to a homomorphic ciphertext decoding operation, that is, for the matrix $U_0$ and $U_1$, homomorphic calculation $z′_k = \frac{1}{N}\left(\overline{U_k}^T \cdot z′ + U_k^T \cdot \overline{z′}\right)$, to obtain two ciphertexts encrypting vectors $z′_0 = \left(t_0, \ldots, t_{\frac{N}{2}-1}\right)$ and $z′_1 = \left(t_{\frac{N}{2}}, \ldots, t_{N-1}\right)$.

$$
U_0 = \begin{bmatrix} 1 & \zeta_0 & \cdots & \zeta_0^{\frac{N}{2}-1} \\ 1 & \zeta_1 & \cdots & \zeta_1^{\frac{N}{2}-1} \\ \vdots & \vdots & \ddots & \vdots \\ 1 & \zeta_{\frac{N}{2}-1} & \cdots & \zeta_{\frac{N}{2}-1}^{\frac{N}{2}-1} \end{bmatrix} \text{ and } U_1 = \begin{bmatrix} \zeta_0^{\frac{N}{2}} & \zeta_0^{\frac{N}{2}+1} & \cdots & \zeta_0^{N-1} \\ \zeta_1^{\frac{N}{2}} & \zeta_1^{\frac{N}{2}+1} & \cdots & \zeta_1^{N-1} \\ \vdots & \vdots & \ddots & \vdots \\ \zeta_{\frac{N}{2}-1}^{\frac{N}{2}} & \zeta_{\frac{N}{2}-1}^{\frac{N}{2}+1} & \cdots & \zeta_{\frac{N}{2}-1}^{N-1} \end{bmatrix} \tag{1}
$$

- *EvalMod*: The goal of *EvalMod* is to homomorphically compute the modular reduction $[\cdot]_q$ function. Since the $[\cdot]_q$ function is not a polynomial function, considering its periodicity, it can be approximated by a sine function to obtain $S(t) = \frac{q}{2\pi}\sin\left(\frac{2\pi t}{q}\right)$. The Taylor polynomial is then used to approximate $S(t)$ and the final approximation polynomial $\frac{q}{2\pi}\sum_{j=0}^{d-1}\frac{(-1)^j}{(2j+1)!}\left(\frac{2\pi t}{q}\right)^{2j+1}$ is obtained. In addition, the double angle formula $\cos(2\theta) = \cos^2\theta - \sin^2\theta$ and $\sin(2\theta) = 2\cos\theta \cdot \sin\theta$ can be used to reduce the calculation cost [25].
- *SlotToCoeff*: *SlotToCoeff* is the inverse process of *CoeffToSlot*, which restores the numbers in the slots to the coefficients of the polynomial. That is, for the given two encoded vectors $z_0 = \left(m_0, \cdots, m_{\frac{N}{2}-1}\right)$ and $z_1 = \left(m_{\frac{N}{2}}, \cdots, m_{N-1}\right)$, the linear transformation $z = U_0 \cdot z_0 + U_1 \cdot z_1$ is computed.

## 4. The Proposed Method

In this section, we introduce the proposed low-degree Hermite neural network (LHDNN), as shown in Figure 2, which includes the LotHps activation layer and weight initialization and regularization modules. In addition, we introduce the variable-weighted difference training (VDT) strategy.

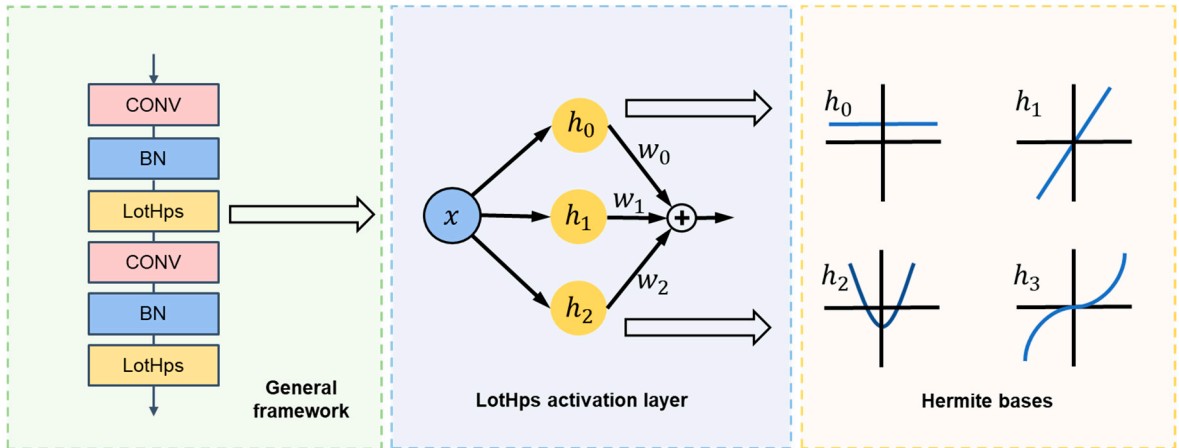

**Figure 2.** Low-degree Hermite neural network (LHDNN) framework.

### 4.1. Low-Order Trainable Hermite Polynomials (LotHps) Activation Layer

As previously mentioned, the ciphertext produced by homomorphic encryption only supports addition and multiplication operations. Therefore, the standard ReLU activation function used in deep neural networks does not work properly in this context. To address this issue, we need to use a homomorphic-friendly polynomial as our activation function. Furthermore, performing a single multiplication on the ciphertext produced by homomorphic encryption is computationally expensive, so we want to minimize the number of multiplication operations. To achieve this, we need to design a low-degree polynomial activation function. In this section, we will discuss the important properties of Hermite orthogonal polynomials and how to use them as our activation layer.

**Hermite polynomials:** The Hermite orthogonal polynomials are defined as $H_n(x) = (-1)^n e^{x^2} \frac{d^n}{dx^n} e^{-x^2}$. They have been widely used in various fields due to their many excellent properties [26]. Here, we only introduce the orthogonality that we use. Specifically, for any two distinct non-negative integers $n$ and $m$, the Hermite polynomial $H_n(x)$ and $H_m(x)$ are orthogonal under the weight function $e^{(-x^2)}$, i.e.:

$$\int_{-\infty}^{\infty} e^{-x^2} H_n(x) H_m(x) dx = \sqrt{\pi} 2^n n! \delta_{nm} \tag{2}$$

where $\delta_{nm} = 1$ when $n = m$, otherwise $\delta_{nm} = 0$. Additionally, the Hermite polynomials satisfy a three-term recurrence relation:

$$\begin{cases} h_0(x) = 1, h_1(x) = 2x \\ h_{n+1}(x) = 2x h_n(x) - 2n h_{n-1}(x), n \geq 2 \end{cases} \tag{3}$$

**LotHps based on Hermite polynomials:** Based on the orthogonality of Hermite polynomials, we constructed a low-degree trainable Hermite polynomials (called LotHps) activation function. In order to maintain low depth of multiplication, we only use the lower degree terms $h_0(x)$, $h_1(x)$, and $h_2(x)$ of the Hermite polynomials. The LotHps function proposed by us can be expressed as:

$$LotHps(x) = w_0 h_0(x) + w_1 h_1(x) + w_2 h_2(x) \tag{4}$$

where $w_0$, $w_1$, and $w_2$ are learnable parameters whose values are adjusted adaptively during neural network training. Specifically, during the backward propagation process in the model, the gradient of the weights in the LotHps activation layer can be derived using the chain rule. Assuming that $\theta$ represents the objective function, the gradient of parameters in the LotHps activation layer are:

$$\frac{\partial \theta}{\partial w_i} = \sum_c \frac{\partial \theta}{\partial H(x_c)} \frac{\partial H(x_c)}{\partial w_i} = \sum_c \frac{\partial \theta}{\partial H(x_c)} h_i(x_c), \ i = 0, 1, 2 \tag{5}$$

where $c$ represents the number of specific input channels, $x_c$ represents the input value of the $c$-th channel of the Hermite activation layer, and $\frac{\partial \theta}{\partial H(x_c)}$ represents the gradient back-propagated from a deeper layer. With the gradient, we can update the values of $w_0$, $w_1$, and $w_2$ through optimization algorithms such as stochastic gradient descent [27] to minimize the loss function.

As to why we use the Hermite polynomials instead of the similar Legendre, Chebyshev and Laguerre polynomials, etc., this is because only the orthogonal interval of Hermite polynomials is $[-\infty, +\infty]$. This means that no matter how large the output value of the batch normalization (BN) layer in a DNN is, it is always in the orthogonal interval of the Hermite polynomial. Although we can satisfy the orthogonality condition for other orthogonal polynomials by scaling the input values, we will undoubtedly introduce more weights to train [26].

### 4.2. Weight Initialization and Regularization Module

**Weight initialization of LotHps:** To reduce the uncertainty of weight random initialization for the LotHps activation layer, we propose a novel weight initialization method that can make the error in the initial stage of model training smaller. Specifically, we use the weight coefficients obtained by approximating the ReLU function as the initial weights of the LotHps activation layer, which provide a good starting point for it. The approximate method we propose is as follows:

Assume that $\varphi_0(x), \varphi_1(x), \cdots, \varphi_n(x)$ is a family of functions with weight orthogonal about the point set $\{x_i\}(i = 0, 1, \cdots, m)$. In this case, we use a family of Hermite orthogonal functions where $\varphi(x)$ refers specifically to $h(x)$. Specify that the approximation function consisting of this family of orthogonal functions takes the form:

$$S(x) = a_0 \varphi_0(x) + a_1 \varphi_1(x) + \cdots + a_n \varphi_n(x) \tag{6}$$

The conventional approximation method is to minimize the sum of squared errors, as shown in the following equation:

$$\|\delta\|_2^2 = \sum_{i=0}^{m} \left[ S^*(x_i) - y_i \right]^2 = \min_{S(x) \in \varphi} \sum_{i=0}^{m} [S(x_i) - y_i]^2 \tag{7}$$

where $S^*(x)$ represents the best approximation polynomial, and $(x_i, y_i)$ represents the sample points, in this case specifically the points on the ReLU function.

The conventional method only provides the best fit for the original function, which is effective for the forward propagation process of the neural network model. However, the gradient of its approximation function may have a large difference with the gradient of the original function, leading to a large error in the backward propagation process of the model. To address this, we consider adding the error of both derivative functions to the objective function. Additionally, since the output values of the batch normalization layer follow a normal distribution, the values are mostly concentrated around 0. We use

a weight function $\sqrt{1-(x/l)^2}$ to better approximate the function values and derivative values around 0, resulting in the final approximation objective we use:

$$\min_{S(x)\in\varphi}\sum_{i=0}^{m}\sqrt{1-(x/l)^2}\cdot[S(x_i)-f(x_i)]^2+[S'(x_i)-f'(x_i)]^2 \tag{8}$$

where $S'(x)$ represents the derivative of the approximating function, $f'(x)$ represents the derivative of the approximated function, and $[-l,l]$ represents the approximation interval.

**Weight regularization of LotHps:** During DNNs model training, we found that significant changes in the values of the weights of the LotHps activation layer or changes in sign caused large fluctuations in the model loss. To prevent the instability of weights during training, we combined the aforementioned weight initialization techniques and proposed a novel weight regularization module that can also improve the generalization ability of the model.

Because $w_i$ corresponds to the Hermite polynomial $h_i(x)$, and $h_i(x)$ itself is not in the same dimension, so we first use $w_i = \frac{1}{\sqrt{2^i i!}}w_i$ to get dimensionless $w_i$. Then, let $W_t = (w_0^t, w_1^t, \cdots, w_2^t)$ represent the weight of the LotHps activation layer during training, and $W_s = (w_0^s, w_1^s, \cdots, w_2^s)$ represent its initial weight. We calculate the relative Euclidean distance between $W_t$ and $W_s$ as a regularization term to constrain $W_t$. Finally, the relative Euclidean distance for a single LotHps activation layer is as follows:

$$d = \frac{\|W_t - W_s\|}{\|W_s\|} \tag{9}$$

The parameters $d$ of each LotHps activation layer of the model are subsequently averaged while multiplying by a regular term parameter $\lambda$ to control the strength of the regularity, in order to obtain the LotHps weight regularity term, which is as follows:

$$\Omega(W_t) = \lambda \cdot \frac{1}{N}\sum_{i=0}^{N}d_i \tag{10}$$

where $N$ denotes the number of LotHps activation layers and $d_i$ represents the relative Euclidean distance for the $i$-th LotHps activation layer.

### 4.3. Variable-Weighted Difference Training (VDT) Strategy

The proposed LotHps activation layer shows good performance on shallower DNNs, but its accuracy degrades slightly when applied to deeper networks. Inspired by knowledge distillation techniques, we propose a variable-weighted difference training (VDT) strategy designed to reduce the difference between DNNs using LotHps functions (called LotHps-based models) and DNNs using ReLU functions (called ReLU-based models), thereby improving the accuracy of LotHps-based models.

In Figure 3, we demonstrate our novel VDT strategy, which leverages the original ReLU-based model as a teacher to supervise the training of the LotHps-based model. The proposed VDT strategy consists of two loss terms: the first loss term corresponds to the activation loss between the LotHps-based model and the ReLU-based model, while the second loss term comprises the output loss of both models and the cross-entropy between the LotHps-based model output and the true label. Notably, we utilize a weight function denoted by $\rho(x)$ in Equation (16) to achieve a smooth transition between the two aforementioned loss terms.

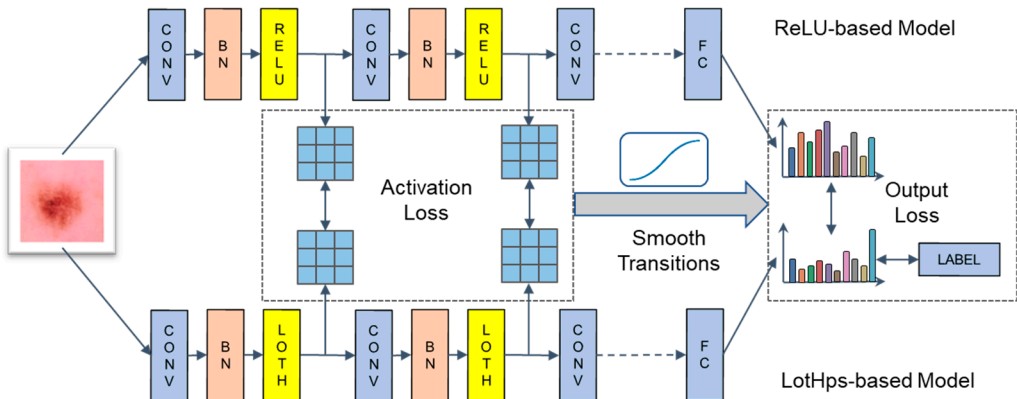

**Figure 3.** Variable-weighted difference training (VDT) schematic.

**Activation Loss:** To quantify the activation loss for the two models, we utilize Kullback–Leibler (KL) divergence. Specifically, we assume that activation layer output distributions of the LotHps and ReLU-based models are represented as $q(x)$ and $p(x)$ respectively. The KL divergence of $q(x)$ and $p(x)$ can be expressed as:

$$D_{KL}(p \parallel q) = \sum_{j=1}^{M} \left[ p(x_j) \log p(x_j) - p(x_j) \log q(x_j) \right] \tag{11}$$

Let $N$ be the number of activation layers, and $D_i(p_i \parallel q_i)$ be the KL divergence of the $i$-th activation layer. Then, the first loss term is:

$$loss\_act = \sum_{i=1}^{N} D_i(p_i \parallel q_i) = \sum_{i=1}^{N} \sum_{j=1}^{M} p_i(x_j) \left[ \log p_i(x_j) - \log q_i(x_j) \right] \tag{12}$$

where $p_i(x)$ and $q_i(x)$ are the $i$-th activation layer output distributions of the LotHps-based model and the ReLU-based model, respectively.

**Output Loss:** For the loss terms of the final output distribution of the two models, we utilize the approach suggested in [28], which uses a response-based knowledge distillation method with a soft target technique:

$$Q^T[i] = \frac{\exp(z_i/T)}{\sum_j \exp(z_j/T)} \tag{13}$$

where $Q^T[i]$ is the soft target version of the class $i$ sample prediction, $T$ represents the temperature, and $z_i$ is the logarithm of the original prediction. The second loss term is obtained by combining the output distribution differences and hard label loss as follows:

$$loss\_out = \alpha_1 T^2 \cdot CE\left(Q_H^T, Q_R^T\right) + \alpha_2 \cdot CE\left(Q_H^1, y_{\text{true}}\right) \tag{14}$$

where $Q_H^T$ and $Q_R^T$ represent the soft target outputs of the LotHps and ReLU-based models, respectively, $y_{true}$ represents the true label, and $CE$ represents the cross-entropy function. $\alpha_1$ and $\alpha_2$ are two hyperparameters that control the relative magnitude of the two cross-entropy losses.

**Loss Smooth Transition:** We did not simply weigh the two aforementioned loss terms arithmetically. Instead, we used a smooth transition function $\rho(x)$ as a dynamic weighting function to achieve a smooth transition from $loss\_act$ to $loss\_out$. In this way, during the initial training, the main goal of the LotHps-based model is to reduce the difference between it and the activation output distribution of the ReLU-based model. In the later stages of

training, the LotHps-based model adjusts the model according to the final learning goals. The specific expression of the smooth transition function $\rho(x)$ is as follows:

$$\rho(x) = \begin{cases} 2\left(\frac{x-e_0}{e_1-e_0}\right)^2, e_0 \leq x \leq c \\ 1 - 2\left(\frac{x-e_1}{e_1-e_0}\right)^2, c < x \leq e_1 \\ 1, x > e_1 \end{cases} \tag{15}$$

where $e_0 = 0$ represents the initial moment of training, $e_1$ is an adjustable parameter representing the complete transition moment, and $c = \frac{1}{2}(e_0 + e_1)$ represents the moment when two loss priorities are equal. Ultimately, the total loss item of the LotHps-based model can be expressed as:

$$Loss = \rho(x) \cdot loss\_act + (1 - \rho(x)) \cdot loss\_out \tag{16}$$

## 5. Implementation Details

The specific implementation is divided into two parts: model training on unencrypted data, and privacy inference on encrypted data. We built three models in three different datasets. The first model is a network (named CNN-6) with five nonlinear activation layers on the MNIST dataset. The second model is an AlexNet model on the Skin-Cancer dataset. The third model is a ResNet-20 model on the CIFAR-10 dataset. Please see Table 1 for more details on the models. In this section, we will present the datasets and models we used, as well as the security parameters and inference optimization methods.

**Table 1.** Model information.

| Dataset | Input Shape | Model | Params (MB) | Mul-Depth |
|---|---|---|---|---|
| MNIST | $1 \times 28 \times 28$ | CNN-6 | 0.57 | 5 |
| Skin-Cancer | $3 \times 32 \times 32$ | AlexNet | 88.74 | 7 |
| CIFAR-10 | $3 \times 32 \times 32$ | ResNet-20 | 1.04 | 19 |

CNN-6 represents [C-B-A-C-B-A-P] $\times$ 2-F-A-F, C represents the convolutional layer, B represents the batch normalization layer, A represents the activation layer, P represents the average pooling layer, and F represents the fully connected layer. Mul-depth represents the multiplication depth of the model.

### 5.1. DataSets

- MNIST [29]: The MNIST dataset consists of single-channel images of 10 handwritten Arabic numerals. It includes 50,000 images in the test set and 10,000 images in the training set, each with a size of $28 \times 28$ pixels. In total, there are 60,000 images in the MNIST dataset.
- Skin-Cancer [30]: The Skin-Cancer dataset consists of medical images of different types of skin cancer, with a total of 10,015 images belonging to seven different categories. We modified the size of all images to $32 \times 32$ pixels, and divided the dataset into a training set and a test set in an 8:2 ratio. Because the data was severely imbalanced, we performed data enhancement and resampling operations on the training data.
- CIFAR-10 [31]: The CIFAR-10 dataset consists of color images of 10 different objects, with a total of 60,000 images. It includes 50,000 images in the test set and 10,000 images in the training set, each with a size of $32 \times 32$ pixels. The training set is extended by random rotation and random clipping.

### 5.2. Model Architecture

For the MNIST dataset, we built a network (named CNN-6) containing four convolutional layers and two fully connected layers. The exact arrangement of the network layers is shown in Table 1, where C represents the convolutional layer, B represents the batch normalization layer, A represents the activation layer, P represents the average pooling layer, and F represents the fully-connected layer. For the Skin-Cancer dataset, we modified

the standard AlexNet network [32] to accommodate the size of the input images, and replaced the maximum pooling layer with a homomorphism-friendly average pooling layer. For the CIFAR-10 dataset, we used the standard ResNet-20 network [33].

### 5.3. Approximation Interval of Weight Initialization

Because the outputs of the intermediate layers of different models are different, this also leads to different input values of the activation layer. If we choose too small an interval for weight initialization, those larger input values, which increase rapidly after activation, will easily lead to gradient explosion. At the same time, if the interval is too large, it will lead to a poor approximation effect of the LotHps function to the ReLU function, resulting in a large initial training loss in the LotHps-based model. Therefore, it is especially important to choose a good initialization interval, and we use the maximum absolute value of each activation layer input as the parameter $l$ of the approximation interval $[-l, l]$. When training the original ReLU-based model, we counted this value as 23.2, 32.8 and 40.3 for the three models, so when training the LotHps-based model, we used these values as the parameter $l$.

### 5.4. Safety Parameter Setting

As with other encryption schemes, the CKKS homomorphic encryption scheme requires parameters to be set to ensure that known attacks are computationally infeasible. We chose different configurations for the different models, and all configurations satisfy 128-bit security, which means that an adversary would need to perform at least $2^{128}$ basic operations to crack the scheme with probability 1. The polynomial degree $N$ of the first configuration is set to $2^{15}$, with integer and fractional partial precision set to 10 and 50, respectively, and a multiplication depth of 14. The second configuration is set to $N = 2^{15}$, with integer and fractional partial precision set to 10 and 39, respectively, and a multiplication depth of 20. The third configuration is set to $N = 2^{16}$, with integer and fractional partial precision set to 12 and 48, respectively, and a multiplication depth of 9. In addition, in the third configuration, we use bootstrapping technology. The degree of the approximation polynomial of the modulus function is set to 14, and the maximum length of the modulus is 1332, which meets 128-bit security, while Lee et al. [14] only meets 111.6 bits of security. The parameters for each configuration are listed in Table 2.

**Table 2.** Safety parameters configuration.

| Model | $\lambda$ | Integer Precision | Fractional Precision | Evaluation Level | Degree | Bootstrapping Level |
|---|---|---|---|---|---|---|
| CNN-6 | 128 | 10 | 50 | 14 | $2^{15}$ | - |
| AlexNet | 128 | 10 | 39 | 20 | $2^{15}$ | - |
| ResNet-20 | 128 | 12 | 48 | 9 | $2^{16}$ | 13 |

### 5.5. Inference Optimization

When performing model inference on ciphertexts, some ciphertext packing techniques and special convolution methods are required to reduce the complexity of the inference process. Aharoni et al. [34] proposed a data structure called the graph block tensor and an interleaved packing method that can effectively reduce the latency and memory consumption of ciphertext inference, and can easily adapt the output of one convolutional layer as the input for the next convolutional layer. The graph block tensor is a data structure that packs tensors into fixed-size blocks according to the requirements of homomorphic encryption and allows them to be manipulated similarly to regular tensors [34]. We use this approach, and the shape and packing of the tensor blocks for the three models are shown in Table 3.

**Table 3.** Parameter setting of inference optimization method.

| Model | Batchsize | Tile Shape | Packing Mode |
|---|---|---|---|
| CNN-6 | 64 | $8 \times 8 \times 1 \times 4 \times 64$ | F-W-H-C-B |
| AlexNet | 8 | $8 \times 4 \times 4 \times 16 \times 8$ | C-W-H-F-B |
| ResNet-20 | 16 | $4 \times 16 \times 16 \times 2 \times 16$ | C-W-H-F-B |

C represents the number of channels, W represents the width of the image, H represents the height of the image, F represents the number of convolution kernels, B represents the batchsize.

Given an image input $I[w_I, h_I, c, b]$, each dimension represents the height and width, number of channels and batch of the image, and the convolution kernel size $F[w_F, h_F, c, f]$; each dimension represents the height and width, number of input channels, and number of output channels of the convolution kernel (i.e. the number of convolution kernels) respectively. The two are multiplied to obtain $O[w, h, c, b, f]$. For a particular block shape $[t_1, t_2, t_3, t_4, t_5]$ and packing method, C-W-H-F-B can be decomposed and rearranged to $O\left[\frac{c}{t_1}, \frac{w}{t_2}, \frac{h}{t_3}, \frac{f}{t_4}, \frac{b}{t_5}\right]$ for $O$.

## 6. Evaluation

### 6.1. Plaintext Training

In this section, we train the proposed LotHps-based model on three different datasets. Please note that our focus is not on achieving higher classification accuracy, but on reducing the accuracy gap between the LotHps-based model and the original ReLU-based model. This ensures that our LotHps-based model can be effectively applied to privacy inference on ciphertexts.

#### 6.1.1. MNIST Dataset

For the MNIST dataset, we evaluated CNN-6 using our proposed methods. We set the batchsize to 64 and used the Adam optimizer. The initial learning rate for both the ReLU and LotHps-based models is set to 0.001. We adopt a dynamic adjustment learning rate strategy, where the learning rate decays by half every 10 epochs of training. The final accuracy of the original ReLU-based model is 99.57%, and the final accuracy of the LotHps-based model is 99.62%. We also tried to increase the number of Hermite orthogonal bases in the activation layer, but found that the accuracy actually decreased and the training process was prone to gradient explosion due to the runge phenomenon of higher degree polynomials. Since our LotHps-based model is more accurate than the original ReLU-based model, we do not need to use the variable-weighted difference training strategy to further improve its performance. This avoids adding additional computational overhead.

We compared the accuracy and errors of the methods proposed in other literature on the MNIST dataset, and the models used by these methods are shown in Table 4. CryptoNets [9] used $x^2$ to replace the original ReLU activation function, resulting in an absolute accuracy reduction of 0.32%. PPCN [10] used the Taylor approximation of the softplus function combined with the BN layer, which results in an accuracy reduction of 0.29%. CryptoDL [12] used a 2-degree polynomial approximation of the derivative function of ReLU and integrates it to obtain a 3-degree polynomial to replace the ReLU function. The original model has an accuracy of 99.67%, but this drops to 99.52% after the replacement. QuaiL [35] used the idea of dynamic programming to train the polynomial as the model of activation function, and its accuracy was 99.26%. We used the approximate polynomial from Approx-ReLU [21] as the activation function, which resulted in a 0.03% reduction in model accuracy. However, the model test accuracy curve of this method fluctuates widely, as shown in Figure 4.

**Table 4.** Contrast of plaintext accuracy on three datasets.

| Dataset | Method | Model | Replacement Accuracy | Original Accuracy | Accuracy Difference |
|---------|--------|-------|---------------------|-------------------|---------------------|
| MNIST | CryptoNets [9] | C-A-P-A-F | 98.95 | 99.28 | −0.33 |
| | PPCN [10] | [C-B-A-C-B-A-P]*3-F-F | 99.30 | 99.59 | −0.29 |
| | CryptoDL [12] | [[C-B-A]*2-P-C-B-A]*2-F-F | 99.52 | 99.56 | −0.04 |
| | Approx-ReLU [21] | [C-B-A-C-B-A-P]*2-F-A-F | 99.54 | 99.57 | −0.03 |
| | QuaiL [35] | LeNet-5 | 99.26 | 99.32 | −0.06 |
| | **LHDNN** | **[C-B-A-C-B-A-P]*2-F-A-F** | **99.62** | **99.57** | **0.05** |
| Cancer | CryptoNets [9] | AlexNet | 67.26 | 81.14 | −13.88 |
| | Approx-ReLU [21] | AlexNet | 75.85 | 81.14 | −5.29 |
| | **LHDNN (no VDT)** | **AlexNet** | **78.08** | **81.14** | **−3.06** |
| | **LHDNN** | **AlexNet** | **81.10** | **81.14** | **−0.04** |
| CIFAR-10 | CryptoDL [12] | CNN-10 | 91.50 | 94.20 | −2.70 |
| | PACN [13] | VGG-16 | 91.87 | 91.99 | −0.12 |
| | QuaiL [35] | VGG-11 | 82.85 | 90.46 | −7.61 |
| | QuaiL [35] | ResNet-18 | 85.72 | 93.21 | −7.49 |
| | BDGM [28] | AlexNet | 87.20 | 90.10 | −2.90 |
| | **LHDNN (no VDT)** | **ResNet-20** | **88.97** | **91.58** | **−2.61** |
| | **LHDNN** | **ResNet-20** | **91.54** | **91.58** | **−0.04** |

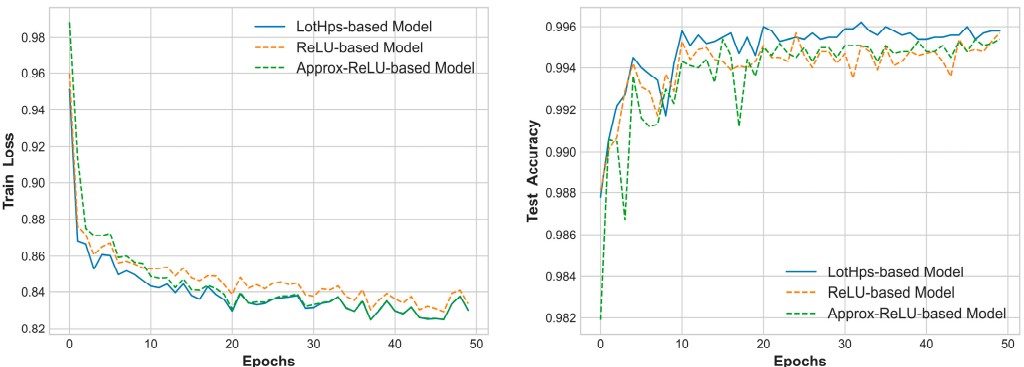

**Figure 4.** Comparison of train loss and test accuracy on MNIST.

Figure 4 shows the training loss and test accuracy curves of the three activation functions: ReLU, LotHps, and Approx-ReLU [21]. From the figure, we can see that the model training losses of our proposed LotHps and Approx-ReLU are very close after 30 epochs. However, the model losses of our method are smaller in the early training period, which indicates the effectiveness of our proposed weight initialization method. Observing the test accuracy curves, we can see that the model test accuracy curves corresponding to the Approx-ReLU fluctuate more, while the curves of the LotHps are more stable and have higher test accuracy. This reflects that our weight regularization module can ensure the stability of model generalization ability in the process of model training and can effectively improve the model's generalization ability.

6.1.2. Skin-Cancer Dataset

For the Skin-Cancer dataset, we evaluated AlexNet using our proposed methods. We set the batchsize to 64, the optimizer to Adam, and the initial learning rate to 0.001. The learning rate decays to 0.5 times the original value every 10 steps, and the LotHps regularization parameter $\lambda$ is set to 0.0005. The accuracy of the original ReLU-based model is 81.14%, while the LotHps-based model without VDT strategy reaches 78.08%, resulting in a 3.06% accuracy reduction. In contrast, the LotHps-based model trained with VDT strategy achieves an accuracy of 81.10%. We set the value of $e_1$ in the smooth transition function $\rho(x)$ to 20, meaning that the model discarded the *loss_act* term after 20 epochs and only used the

reduction of *loss_out* as the learning goal. We also tried making $e_1$ fluctuate up and down by 5 steps and found that the accuracy fluctuates within 0.03%. Additionally, we evaluated AlexNet using two methods: the squared activation function [9] and Approx-ReLU [21], with a final accuracy of 67.26% and 75.85%, respectively. This is a significant difference in accuracy compared to the original ReLU-based model.

### 6.1.3. CIFAR-10 Dataset

For the CIFAR-10 dataset, we evaluated ResNet-20 using our proposed method. The original ReLU-based model using the training hyperparameters from literature [33] achieved an accuracy of 91.58%. For our LotHps-based model, the optimizer was Adam, the LotHps regularization parameter $\lambda$ was set to 0.0005, and the initial learning rate was set to 0.001. The learning rate was dynamically adjusted, decreasing by a factor of 0.5 every 10 epochs. Without using the VDT strategy, our LotHps-based model achieved an accuracy of 88.97%. Using the VDT strategy, our LotHps-based model achieved an accuracy of 91.54%, an improvement of approximately 2.57%, with the weighting function parameter $e_1$ set to 25. We also observed a final accuracy fluctuation within 0.05% by fluctuating $e_1$ by 5 steps.

We compared the accuracy and errors of the methods proposed in other literature to the CIFAR-10 dataset, and the models used by these methods are shown in Table 4. Using the Approx-ReLU [21] function as the activation layer of ResNet-20, it achieved an accuracy of only 62.38%. CryptoDL [12] used its proposed method to evaluate the 10-layer DNN (called CNN-10) and achieved an accuracy of 91.50%, but the accuracy was reduced by 3.7% compared to the original ReLU-based model. QuaiL [35] used its proposed method to evaluate VGG-11, and its accuracy was reduced by about 7.61%, indicating that the proposed method was not applicable to the depth model. Lee et al. [13] used a 27-degree polynomial instead of the ReLU function to evaluate the VGG-16 model, resulting in a 0.11% reduction in accuracy. However, their high-degree polynomial caused a significant delay in cryptographic inference. BDGM [28] proposed a smooth replacement activation function method and evaluated AlexNet combined with a knowledge distillation technique. The replacement accuracy was 87.20%, while the original ReLU-based model's accuracy was 90.10%.

### 6.2. Ciphertext Inference

In this subsection, we present the results of our trained model's ability to reason over ciphertexts. We implemented our proposed model using the Helayers library [34], released by IBM Research, which connects to the three underlying homomorphic cryptographic libraries—HELib, HEAAN, and SEAL. Our three models all use the HEAAN ciphertext, and our simulation environment is a cloud server with an Intel(R) Xeon(R) Gold 5218 CPU @ 2.30 GHz processor (48 cores) and 100 GB of RAM.

### 6.2.1. Analysis and Comparison of Ciphertext Inference Results

In most cases, the cloud would perform privacy classification on multiple images from the user, so batch inference can be performed on the images. However, due to memory limitations on the server we use, we cannot set the batch size too large. At the same time, it cannot be too small either, as this would increase the amortized runtime. Taking these constraints into account, we set the inference batchsize for CNN-6, AlexNet, and ResNet-20 to 64, 8, and 16 respectively.

Table 5 shows the specific time consumption of the three model reasoning processes using an RNS-CKKS encryption scheme. We chose HEAAN as the back-end encryption library and only encoded the model parameters without encrypting them. For time consumption, the total reasoning times of our CNN-6, ALexNet, and ResNet-20 are 142.62 s, 244.90 s, and 1027.96 s, respectively, and the corresponding amortized running times (time per image) are 2.23 s, 30.61 s, and 64.25 s, respectively. In practical applications, we can run our model on GPU to further reduce inference latency [36]. Regarding memory usage,

the total reasoning memory usages of the three are about 23 G, 81 G, and 96 G respectively. As can be seen from the table, the AlexNet model parameter encoding stage has the largest memory occupation, which is caused by the huge number of parameters in its full connection layer. For the ResNet-20 model, the polynomial degree corresponding to CIFAR-10 image ciphertext is $N = 2^{16}$, and bootstrapping refresh noise is adopted, so the corresponding context memory is relatively large. We also evaluated the memory consumption of inferencing only one image at a time, with the three models consuming 6.91 G, 24.58 G, and 34.63 G of memory, respectively.

**Table 5.** Ciphertext inference time (s) and memory usage (GB) consumption.

| Option | Dataset | Model | Initialization Context | Model Encoding | Input Encryption | Output Decryption | Inference |
|--------|---------|-------|------------------------|----------------|------------------|-------------------|-----------|
| Time | MNIST (b = 64) | CNN-6 | 1.39 | 0.11 | 0.41 | 0.0045 | 142.62 |
| | Cancer (b = 8) | AlexNet | 2.97 | 2.70 | 0.75 | 0.0049 | 244.90 |
| | CIFAR-10 (b = 16) | ResNet-20 | 8.18 | 0.53 | 0.18 | 0.0233 | 1027.96 |
| Memory | MNIST (b = 64) | CNN-6 | 1.67 | 0.0021 | 2.26 | 0.0005 | 19.12 |
| | Cancer (b = 8) | AlexNet | 5.13 | 0.1735 | 3.94 | 0.0005 | 71.91 |
| | CIFAR-10 (b = 16) | ResNet-20 | 11.62 | 0.0035 | 1.05 | 0.0002 | 83.13 |

We compared our results with existing solutions, and Table 6 presents the results of evaluating DNN using homomorphic encryption on different datasets in previous work. Considering evaluation metrics such as accuracy, inference time, and security, our solution outperforms all other solutions. For example, our MNIST CNN-6 has the highest inference accuracy and the lowest amortized time, achieving a standard 128-bit security. Our SKIN AlexNet has an amortized time of 30.61 s, which is 1.49 times faster than AlexNet using Apprx-ReLU [21], and has higher accuracy. Finally, our CIFAR10 ResNet-20 is 348.20 times faster, 181.88 times faster, and 165.01 times faster than FCryptoNets [37], CryptoDL [12], and PACN [14], respectively, with higher security. Although the amortized time of LDCN [38] is also low, it is an improvement of PACN by its authors, mainly focusing on improving the speed of convolution and bootstrapping. In contrast, we mainly focus on reducing the inference latency of the activation layer, and our ResNet-20 trained using the proposed method can also adopt the convolution and bootstrapping strategy of LDCN [38] to further improve the inference speed.

**Table 6.** Comparison of ciphertext inference results.

| Dataset | Method | Model | Accuracy | Amortized Time | Memory Usage | $\lambda$ |
|---------|--------|-------|----------|----------------|--------------|-----------|
| MNIST | CryptoNets [9] | CNN-4 | 98.95 | 249.6 | N/A | 80 |
| | FCryptoNets [37] | CNN-4 | 98.71 | 39.1 | N/A | 128 |
| | CryptoDL [12] | CNN-4 | 99.25 | 148.9 | N/A | 80 |
| | HCNN-CPU [39] | CNN-3 | 99.00 | 90.32 | N/A | 76 |
| | **LHDNN** | **CNN-6** | **99.62** | **2.23** | **6.91G** | **128** |
| Cancer | Approx-ReLU [21] | AlexNet | 75.85 | 45.68 | 32.89G | 128 |
| | **LHDNN** | **AlexNet** | **81.10** | **30.61** | **24.58G** | **128** |
| CIFAR-10 | FCryptoNets [37] | CNN-8 | 75.99 | 22372 | N/A | 128 |
| | CryptoDL [12] | CNN-9 | 91.50 | 11686 | N/A | 80 |
| | PACN [14] | ResNet-20 | 92.43 ± 2.65 | 10602 | 172G | 111.6 |
| | LDCN [38] | ResNet-20 | 91.31 | 79.46 | N/A | 128 |
| | **LHDNN** | **ResNet-20** | **92.28 ± 2.58** | **64.25** | **34.63G** | **128** |

N/A indicates that this value is not given in the reference.

We note that the results reported in Table 6 were obtained on different machines. The MNIST CNN-4 of CryptoNets [9] was run on a machine with an Intel Xeon E5-1620 CPU at

3.5 GHz with 16 GB RAM. The MNIST CNN-4 of FCryptoNets was run on a machine with an Intel Core i7-5930K CPU at 3.5GHz with 48 GB RAM, while its CIFAR-10 CNN-8 was run on an n1-megamem-96 instance on the Google Cloud Platform, with 96 Intel Skylake 2.0 GHz vCPUs and 1433.6 GB RAM. The CPU evaluation of HCNN [39] was conducted on a machine with an Intel Xeon Platinum CPU at 2.10 GHz with 187.5 GB RAM. The experimental environment for CryptoDL was a machine with Intel Xeon E5-2640 at 2.4 GHz with 16GB RAM. The experimental environment for PACN was a machine with dual Intel Xeon Platinum 8280 CPU (112 cores) with 512GB RAM. The experimental environment for LDCN was a machine with AMD Ryzen Threadripper PRO 3995WX at 2.096 GHz (64 cores) with 512 GB RAM.

### 6.2.2. Analysis of Decryption Error

For the MNIST and SKIN data sets, we inferred ciphertext for the entire test set with 99.62% and 81.10% accuracy, respectively, which is exactly the same as plaintext accuracy. For the CIFAR-10 dataset, we conducted five random samplings of the test set and inferred 384 images each time, with an accuracy of $92.28 \pm 2.58\%$ slightly lower than the plaintext accuracy of $92.59\% \pm 2.27\%$. For this result, we further analyzed the reason. By calculating the relative error between the ciphertext prediction vector and the plaintext prediction vector (called decryption error), we found that the mean decryption error of CNN-6 on the MNIST data set and AlexNet on the SKIN data set were both small, 3 and 4, respectively. However, the ResNet-20 mean relative error on the CIFAR-10 dataset is about 10, which is very large compared to the first two. This is because ResNet-20 uses bootstrapping technology in the ciphertext reasoning process, while bootstrapping in the HEAAN library is approximate, and each bootstrapping will introduce a new error.

In addition, we also calculated the decryption errors of different categories in the dataset and drew box plots by randomly selecting 20 sampling points in each category, as shown in Figure 5. We tested the differences between decryption errors of different categories. For the MNIST dataset, we first tested the homogeneity of variance of decryption errors for all categories, and the result of the $p$-value was $0.596 > 0.05$. The initial hypothesis was accepted, indicating that the decryption errors of different categories were considered to satisfy the homogeneity of variance. Then, one-way ANOVA was performed on the decryption errors, and the result of the $p$-value was $0.580 > 0.05$, indicating that there was no significant difference in decryption errors among different categories in the MNIST dataset. For the SKIN and CIFAR-10 datasets, the $p$-values for the homogeneity of variance test were 0.174 and 0.830, respectively, and the $p$-values for one-way ANOVA on variance were 0.325 and 0.376, respectively. Therefore, the same conclusion as the MNIST dataset was drawn, indicating that there was no significant difference in decryption errors between different categories.

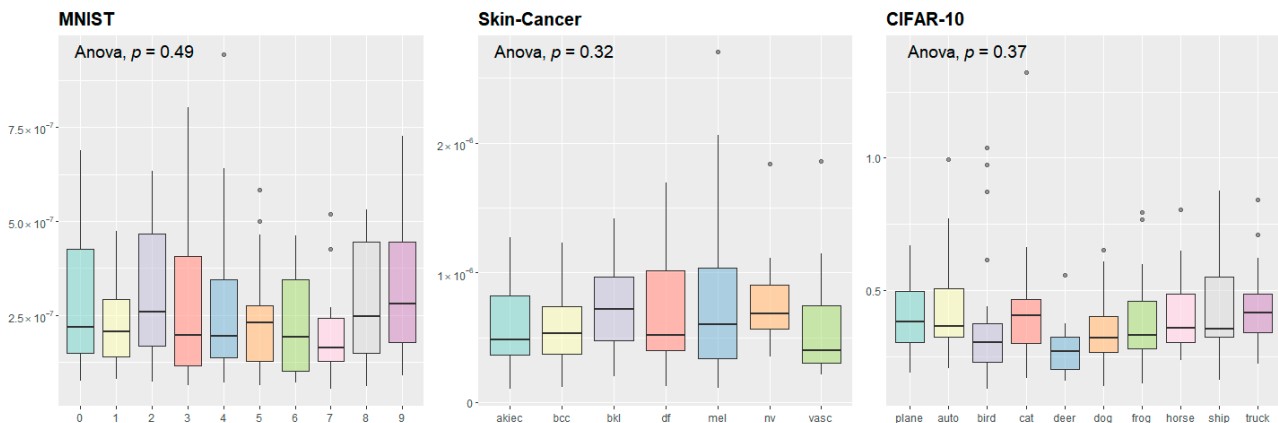

**Figure 5.** Decryption error statistics of different categories.

## 7. Conclusions

In this paper, we propose LHDNN to achieve both high inference accuracy and low inference latency of Deep Neural Networks (DNNs) on homomorphic encrypted data. The LHDNN employs a set of low-degree trainable Hermite polynomials known as LotHps as activation layers in DNNs. Additionally, we integrate a new weight initialization and regularization module into the LotHps activation layer to make the training process of DNNs more stable and strengthen generalization ability. To further enhance the model's accuracy, we propose a variable-weighted difference training (VDT) strategy that uses ReLU-based models to guide the training of LotHps-based models. Extensive experiments conducted on multiple benchmark datasets demonstrate that the LHDNN approach is superior to other methods in terms of both inference speed and accuracy on homomorphic encrypted data.

**Author Contributions:** Methodology, J.Q.; software, J.Q., H.Z. and X.M.; validation, P.Z.; writing—original draft preparation, J.Q. and H.Z.; writing—review and editing, P.Z.; visualization, X.M.; funding acquisition, M.L. and J.W. All authors have read and agreed to the published version of the manuscript.

**Funding:** This research was supported by National Natural Science Foundation of China (No. 62102134), Key Scientific Research Project in Colleges and Universities of Henan Province of China (No. 23A520046 and 23A413005) and Key Science and Technology Project of Henan Province of China (No. 232102210130 and 232102210138).

**Institutional Review Board Statement:** Not applicable.

**Informed Consent Statement:** Not applicable.

**Data Availability Statement:** Not applicable.

**Conflicts of Interest:** The authors declare no conflict of interest.

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
