# Peer review of "LHDNN: Maintaining High Precision and Low Latency Inference of Deep Neural Networks on Encrypted Data"

_applsci, doi:10.3390/app13084815_

Round 1

Reviewer 1 Report

Existing homomorphic encryption schemes support only additive and multiplicative operations and do not support nonlinear activation functions which are commonly used in deep neural networks. Also, the existing schemes uses approximate polynomials instead of standard ReLU activation functions and these approaches suffer from low model accuracy and high inference latency. The proposed framework contains an activation layer which consists of a set of low-degree Hermite orthogonal bases whose combined weights are trainable.  In addition, authors also propose a variable-weighted difference training strategy to avoid accuracy degradation when applying Hermite activation layers to deeper models

Comments:

Title: Title should be properly reframed.

ABSTRACT:

§  Kindly rewrite the abstract highlighting objective, contribution of the paper and proposed outcome of the work.

§  Specify 5-6 proper Index terms.

1.      INTRODUCTION:

·        Authors lacks in presenting the approach. It is difficult and confusing in most of the part in the article.

·        There must be a separate section that provides the information about traditional approaches of Hermite deep neural network.

·        Kindly discuss What is low degree Hermite deep neural network in the introduction part.

·        I want a clarity on how Homomorphic encryption works in Hermite deep neural network? How you map these mathematically.?

2.      Related Work:

Some previous works (Recent works in this area) about this research area should also be reviewed

Authors lacks in presenting the approach.

Page No 3 Line no-137: Hermite activation layer in our framework is adaptive? Kindly justify with valid proof.

3.      Mathematical Preliminaries:

Explore clearly how Bootstrapping of CKKS works?

To make the preliminary more clarity I request authors to explain the following along with case study using mathematical proof..

For the polynomial approximation of the residue-taking function, the residue-taking function is first approximated using the sine series, then the Taylor series is used to approximate the sine function in a small range, and the multiplicative angle complex exponential formula is  used to reduce the computational complexity.

4.      Method:.

Need precise and neat use case explanation of different sub sections specified in this section indicating the entire process should be explored.

5.      Results and analysis

a.      The authors are suggested to add more discussions about the comparisons with existing works to show the superiority of the proposed work.

6.      CONCLUSION

                                                    i.          Kindly rewrite the conclusion highlighting the objective, contribution and the results achieved in the article

7.      REFERENCES

                                                    i.          Authors suggested to add more recent citation related to this topic.

Author Response

Thank you very much for taking the time to review this article and for your valuable comments.  Please refer to the attachment for the reply to your review comments.

Reviewer 2 Report

Please add some latest works in the reference section and compare the proposed methods with those works.

Author Response

(The authors gave the same response as above.)

Reviewer 3 Report

The topic is interesting and the paper is well organized. However, the authors should address the following comments:

1.  Please give a paragraph summarizes the limitations of current algorithms and methods at the last of the related work.

2.  Big O Notation is a tool used to describe the time complexity of algorithms. Please describe the time complexity in Big O Notation.

3.  Statistical analysis in terms of p-value for obtained results against the results of some recent related is required.

Author Response

(The authors gave the same response as above.)

Reviewer 4 Report

Excellent work, but for the performance measures (Ciphertext Inference Latency, Table7, and Memory Usage, Table 8), why did not you compare with others' results? This crucial for evaluating the effectiveness of your approach. You can compare to published results (and mention the specifications of the machines they used), it does not have to be on the same machines. It would give the readers an idea of the relative performance.

Author Response

(The authors gave the same response as above.)

Round 2

Reviewer 2 Report

Looks great.